# The Influence of Organically Modified Derivatives of Silica on the Structure and the Wetting Angle Values of Silica Coatings

Anna Adamczyk 

Faculty of Materials Science and Ceramics, AGH University of Science and Technology, Al. Mickiewicza 30, 30-059 Kraków, Poland; aadamcz@agh.edu.pl

**Abstract:** The surrounding environment often acts in a destructive way on materials we apply in our everyday life. The best way to protect them against such activity is to cover the basic materials with coatings possessing different properties, tailored to their applications. Anticorrosive layers are one of the biggest groups of such protective coatings, especially those containing silica or its derivatives. Depending on a type of silica precursor and a method of deposition, one can obtain coatings of different structures and properties. In this work, three different silica precursors were applied: TEOS (tetraethylorthosilane), DDS (dimethyldiethoxysilane) and Aerosil™ (the powder silica). Sols of different concentrations of the aforementioned precursors as well as commercially available preparations (Sarsil H1 4/2 and SILOXAN W290) were applied for thin films deposition by a dip coating or an infiltration method. The substrates could be divided in two groups: metallic (steel and titanium or titanium alloys) and porous (represented by old brick, sandstone and limestone). Following the deposition process, the layers on metallic substrates were additionally annealed at 500 °C to improve their adhesion and mechanical properties, while those on porous materials were dried in air. All prepared coatings were primarily studied by FTIR spectroscopy and X-ray diffraction. The morphology of their surfaces was imaged by SEM and AFM microscopies, which also allowed determination of the roughness of obtained materials. The measurements of wetting angle values enabled to find the relationship between the surface topography, the type of silica precursor and the hydrophobic/hydrophilic properties of the samples. The results confirmed the hydrophobic properties of coatings obtained by the infiltration technique on the porous materials and the high hydrophilicity of the annealed thin film deposited on the metallic substrates.

**Keywords:** anticorrosive thin films; silica precursors; sol-gel method; IR spectroscopy; X-ray diffraction; SEM microscopy; AFM microscopy; hydrophobic/hydrophilic properties

## 1. Introduction

People are actively trying to counteract the destructive influence of the surrounding environment on various types of materials used in everyday life. One of the main purposes of this activity is protection against corrosion caused by various factors. Processes such as the weathering of materials or the rusting of steel cause the destruction of natural building materials, entire buildings or monuments, and often valuable historical heritage [1–4].

To minimize these negative effects of corrosion in the broadest sense, a wide range of different methods is applied. One is to cover the surfaces of materials with different types of coatings. Layers based on the silica matrix allow for changes in their properties by the addition of other components, e.g., $Al_2O_3$, $TiO_2$, $ZrO_2$ or Ag [5–11], and usually provide good mechanical features, good durability and often high chemical resistance. There are several methods of thin films deposition, depending on the type of substrate and the source of the coating components, such as CVD (Chemical Vapor Deposition), PVD (Physical Vapor Deposition), laser ablation [12] or dip-coating. In this work, all coatings were synthesized by the sol-gel method using different $SiO_2$ precursors, focusing on those containing methyl groups, which can be responsible for hydrophobic properties

of materials. The measurement of wetting angle values still remains the simplest method to estimate the hydrophobic or hydrophilic character of the obtained thin films, [13] which is notably important if one would like to protect materials located outside, such as parts of buildings, fences or sculptures.

This work is focused on the research of the structure and morphology of coatings obtained by the sol-gel method on two different types of substrate: porous (represented by ceramics and natural stones) and metallic (represented by steel, titanium and titanium alloys). For all coatings, attempts at measuring the wetting angle values were undertaken to define the hydrophobic or hydrophilic character of synthesized samples.

## 2. Materials and Methods

All thin films were realized by the deposition of sols synthesized by the sol-gel method and application of one of two techniques, dip coating or infiltration, depending on the type of substrate. Substrates were be divided into two groups: metallic ones, represented by steel, titanium and titanium alloy, and porous backgrounds, which were represented by old brick, limestone and two types of sandstone (Figure 1). All samples were thoroughly cleaned or prepared under special conditions before the deposition process. The metallic samples were polished with sand paper and then held for 30 min in acetone in an ultrasonic cleaner. After that, samples were held for 30 s in royal water, then thoroughly rinsed with distilled water and dried. Rock samples were cut from the inside of larger blocks of rock to obtain clean sample surfaces.

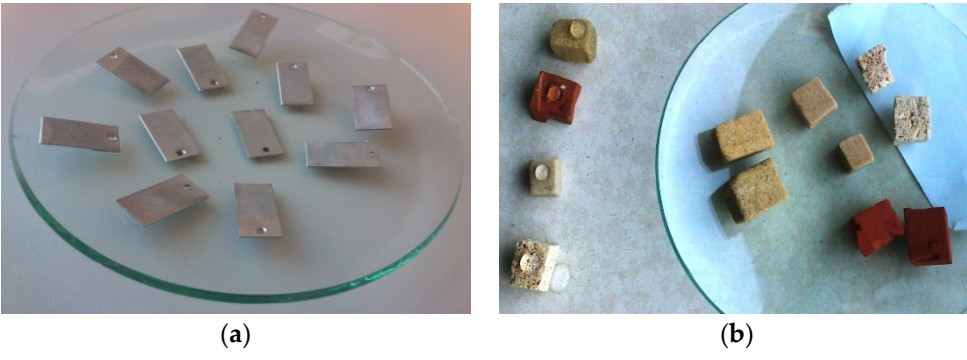

(**a**)           (**b**)

**Figure 1.** Two different types of substrates: (**a**) metallic samples; (**b**) porous samples.

At the beginning, two sols from the group of Ormosils and Ormocers were prepared, based on $SiO_2$ precursors as TEOS (tetraethoxysilane $Si(OC_2H_5)_4$ (Fluka 98%, St. Gallen, Switzerland)) and DDS (dimethyldiethoxysilane $(CH_3)_2(C_2H_5O)_2Si$ (Aldrich 97%, St. Louis, MO, USA)) [14,15]. Then, three silica sols of different $SiO_2$ concentration were also prepared, with TEOS—$Si(OC_2H_5)_4$, DDS—$(CH_3)_2(C_2H_5O)_2Si$ and Aerosil$^{TM}$ (powder silica) as $SiO_2$ precursors. Sols containing only one precursor (TEOS) were prepared in a way to obtain $SiO_2$ concentration from 5% to 14% weight, while in sols containing both TEOS and DDS, the stable molar ratio of TEOS:DDS = 1:1 was kept [16].

For the 5% $SiO_2$ sol, a solution of $NH_4OH$ in $H_2O$ was added to a 1M solution of TEOS in 98% ethanol. The pH was kept at about 8, and the sol was stirred for 2 h and 30 min [17]. To obtain a 10% solution of $SiO_2$, HCl was applied as a catalyst, introduced into the sol at a molar ratio of TEOS:HCl = 1:0.025. A solution of TEOS in 98% ethanol with sufficient HCl added was then stirred for 2 h 30 min [18]. For preparing 14% $SiO_2$ sol, $NH_4OH$ was again used as a catalyst. In the first step, a solution of ethanol with redistilled $H_2O$ and $NH_4OH$ was obtained at the molar ratio $NH_4OH:Si(OC_2H_5)_4$ = 1:100, into which TEOS was slowly dropped at the ratio of TEOS:$Si(OC_2H_5)_4$ = 1:1. The sol was then stirred for 2 h and 30 min on a magnetic stirrer (IKA$^®$ Poland Sp. z o.o., Warszawa, Poland) [19].

For comparison, such commercially available preparations as SARSIL H/14/2 9 [20,21] and SILOXAN W290 [22,23], both containing different modified derivatives of silica, not listed in detail, were selected and then applied as solutions for infiltration.

The coating deposition was carried out by two methods: dip coating and infiltration. During the dip coating process, samples were immersed in the prepared sol, kept in it for 30 s and then pulled out at a constant speed of 4 cm/min. This procedure was repeated twice with a 15 min break, then after 24 h of drying under ambient conditions, thin films were annealed at 500 °C in air (in case of steel substrates) or in argon (when titanium or its alloys were applied as substrate) to improve their adhesion and mechanical properties. Such cycle was performed three times, resulting in a compact thin film consisting of six layers.

The infiltration process involved the immersing of each sample in the selected sol or in the commercial preparation for 30 min and after that time leaving it to dry under ambient conditions.

For all samples, FTIR spectra of samples were collected in a Bruker 70V IR spectrometer (Bruker, Billerica, MA, USA), using the KBr pellets technique for layers scraped off the sample surfaces. In case of selected samples deposited on metallic substrates, the reflection technique of measurements was also applied. The XRD diffraction patterns were measured in an X'Pert diffractometer (Panalytical, Almelo, The Netherlands) using CuKα radiation in standard and GID (Grazing Incidence Diffraction) configurations. The images of the sample surfaces were obtained with a NOVA NANO SEM 200 microscope (FEI Europe Company, Eindhoven, The Netherlands) together with a Genesis XM X-ray microanalysis system (EDAX, Tilburg, The Netherlands), applying the accelerated voltage of 10–18 kV. The nanoscaled images of the samples were obtained with a Multimode 8 AFM Bruker microscope (Bruker Nano Surfaces Division, Santa Barbara, CA, USA) applying Peak Force Tapping mode and SNL10 probes. The wetting angle values were measured by applying a DSA 10 KRÜSS goniometer (KRÜSS GmbH, Hamburg, Germany) to determinate the hydrophilic or hydrophobic nature of samples.

## 3. Results and Discussion

### 3.1. FTIR Studies of Samples

The spectra of the infiltrated and non-infiltrated mineral and ceramic samples presented an excellent set of bands because of the rich chemical composition of natural stones and ceramics (Figures 2 and 3). The most interesting bands, however, are those assigned to the vibrations of $CH_3$ groups, which can be responsible, as mentioned before [13], for the hydrophobic properties of the material. In the analyzed FTIR spectra, bands due to methyl group vibrations were located in the range of 1250–1290 $cm^{-1}$ (the most characteristic and symmetric stretching bands); in the case of Si–$CH_3$ bonds, at 2904 and 2962 $cm^{-1}$ (CH asymmetric and symmetric stretching vibrations, respectively); at 1420 $cm^{-1}$ (CH asymmetric stretching); and finally at 830 $cm^{-1}$ (SiC symmetric bending vibration). As an example, two sets of spectra of old brick and limestone infiltrated with different sols and preparations are shown (Figures 2 and 3). They were selected because, as substrates, they contained the lowest amount of silica (in comparison with, e.g., sandstone). In the spectra of the brick, bands due to the vibrations of Si–O bonds at about 456, 796, 900 and 1088 $cm^{-1}$ are primarily observed, which can be connected with the presence of different silicate and aluminosilicate phases in this sample [24]. Among them, this typical $CH_3$ group vibration band at about 1250–1290 $cm^{-1}$ can be identified. Unfortunately, it is located in a similar range as the band (or shoulder) due to Si–O–Si bond vibrations [25], which is why the band at a similar position can also be observed in non-infiltrated samples. However, the intensity of the aforementioned band changes in the spectra of porous infiltrated samples (Figure 2), which indicates its origin as being due to the $CH_3$ group vibrations more than to the vibrations of Si–O bonds. A similar situation is shown in the spectra of the limestone (Figure 3), where bands connected with the presence of the $CaCO_3$ phase, at 706, 875 and at about 1394–1452 $cm^{-1}$ [26], are first observed. Among them, the band of changing intensity at about 1256–1287 $cm^{-1}$ due to $CH_3$ group vibration can also be identified.

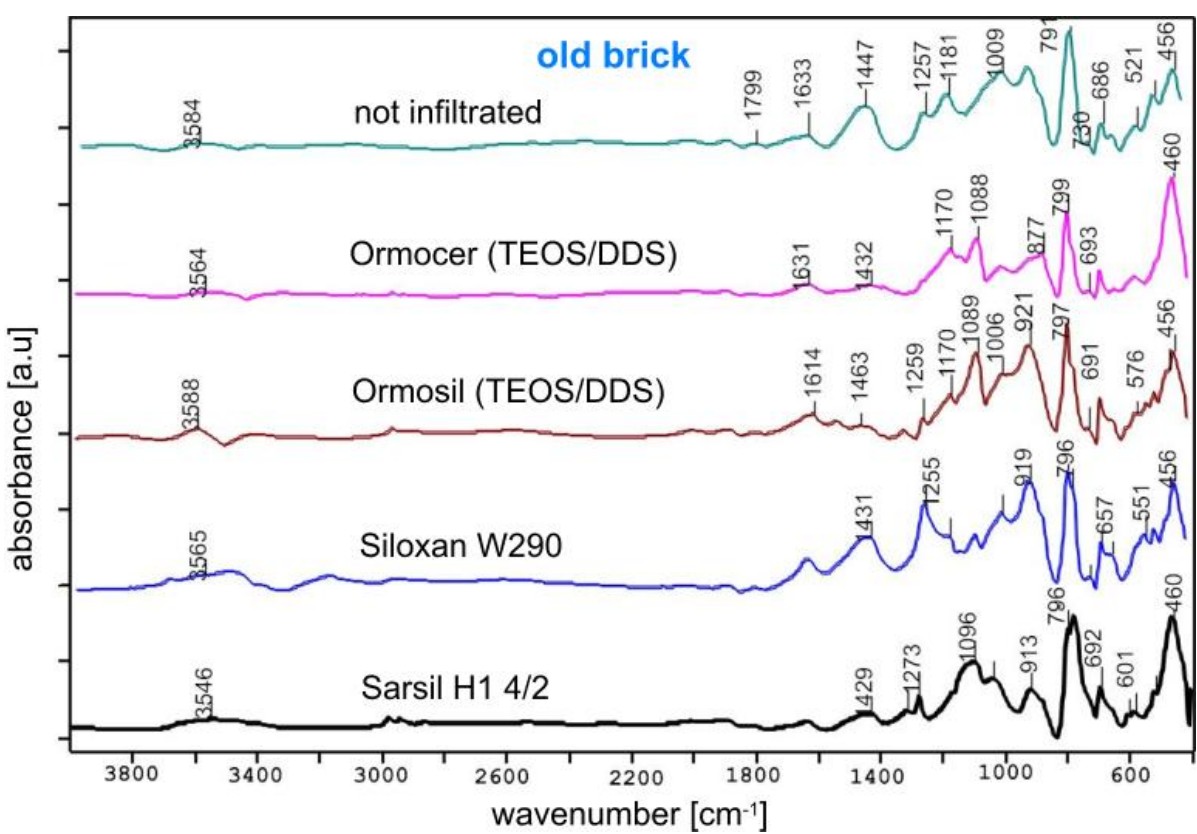

**Figure 2.** FTIR spectra of old brick.

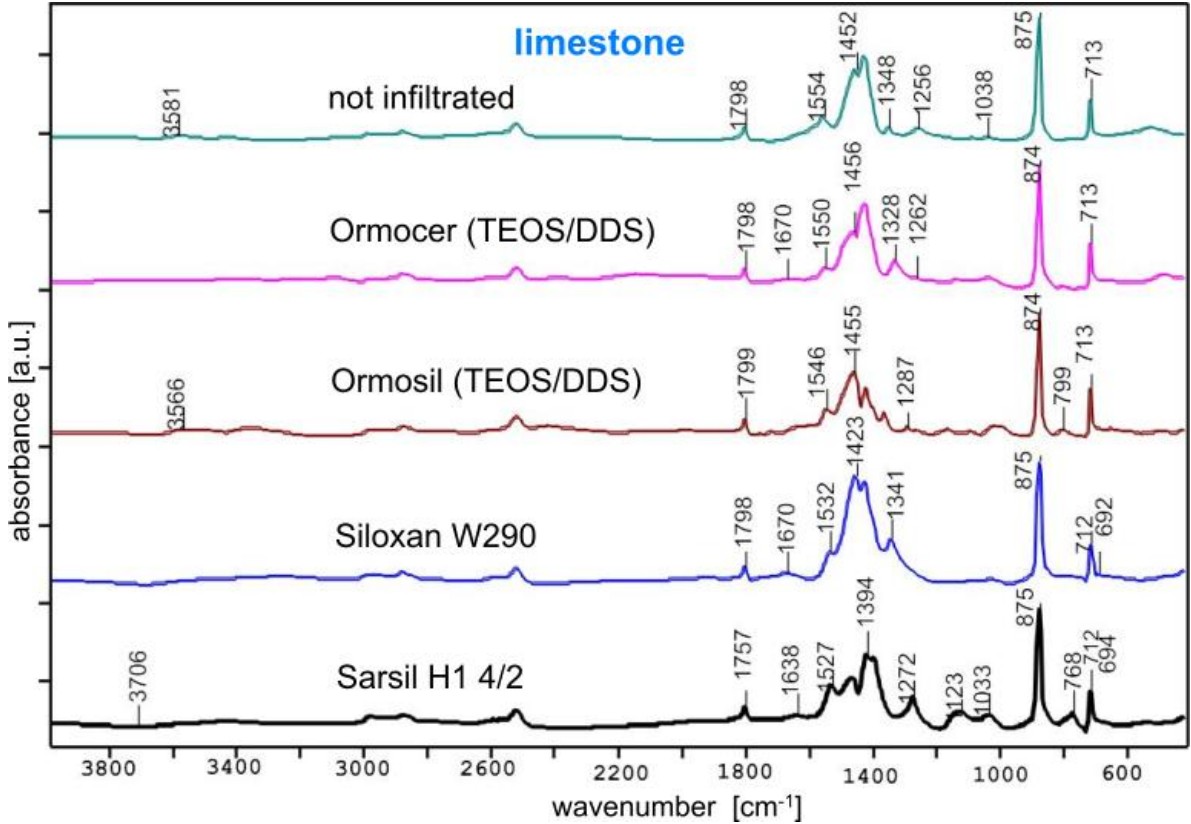

**Figure 3.** FTIR spectra of the limestone.

In the FTIR spectra of the annealed silica coatings deposited on metallic substrates (Figures 4 and 5), most of bands due to Si–O bond vibrations (mentioned earlier during the analysis of the brick and the limestone spectra) are observed, with only one exception of the spectrum of the thin film synthesized from sol containing TEOS and DDS in a 1:1 molar ratio. In this spectrum (Figure 4), the small but sharp and distinct band at 1298 cm$^{-1}$ assigned to CH$_3$ group vibration can be noticed, while bands due to Si–O bond vibrations at about 1220–1234 cm$^{-1}$ are observed as wide bands or shoulders. Such broad banding is due probably to Si–O bond vibrations and is observed in the spectrum of the coating obtained with only TEOS-containing sol (Figure 5). The presence of the band typical of CH$_3$ molecule vibrations in the earlier mentioned spectrum (Figure 5) is somewhat surprising due to the prior annealing of thin films on metallic substrates at 500 °C; it indicates that some of the methyl groups remain incorporated into the layers' structure and can thus influence the hydrophobicity of samples.

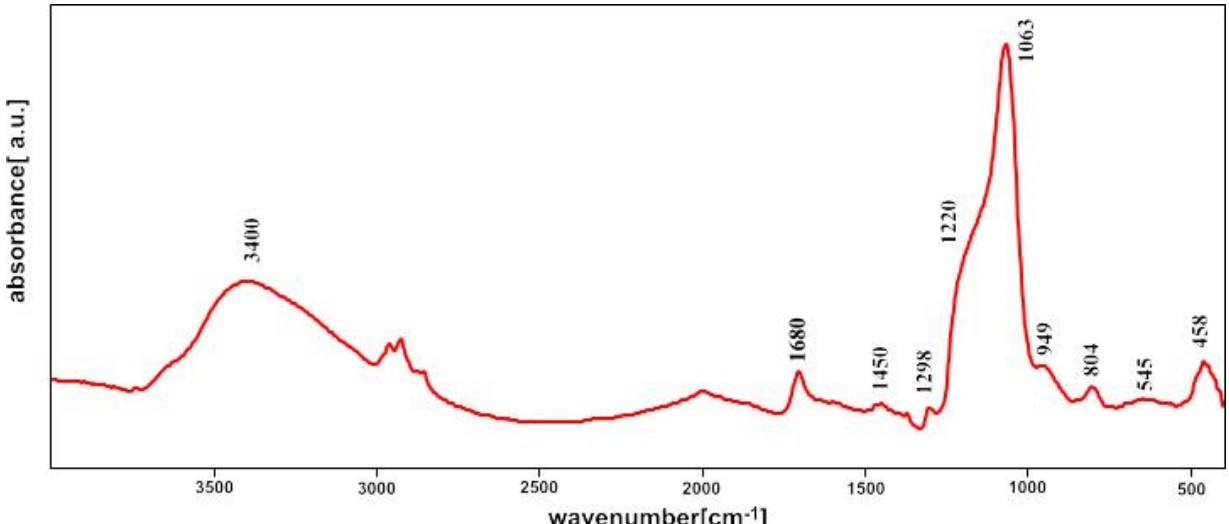

**Figure 4.** FTIR spectrum of the thin film prepared from the sol containing TEOS and DDS and deposited on steel substrate.

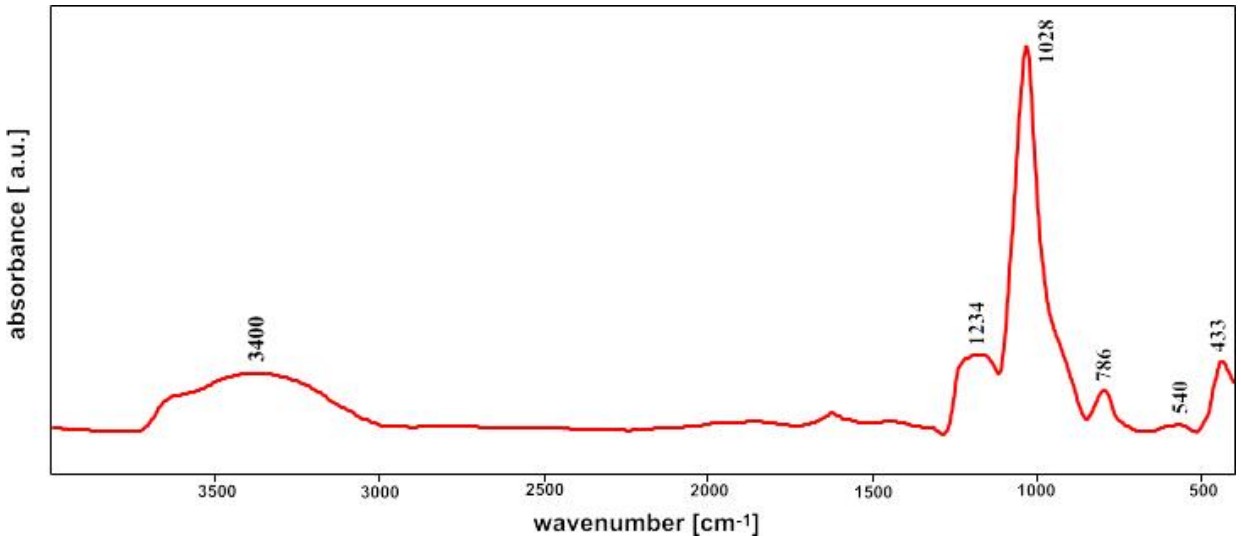

**Figure 5.** FTIR spectrum of the thin film synthesized from the basic containing TEOS sol (5% mas. SiO$_2$) and deposited on steel substrate.

FTIR studies confirm the presence of methyl groups in all coated samples, irrespective of the method of deposition from sols containing methyl groups. Thus, one can study the relations between the influence of the methyl groups' presence and the morphology of the coatings' surfaces on the potential hydrophobicity or hydrophilicity of samples.

### 3.2. XRD Studies of Samples

XRD diffraction patterns (not included in this work) primarily show reflections assigned to phases present in the substrates; thus, one can identify anatase and rutile phases in the case of titanium alloys as background or austenite together with other types of Fe-Cr phases for the steel substrate. The diffraction patterns of coatings deposited on the natural stones show peaks due to calcite, with the small addition of pyrophilite in the case of the limestone sample and of quartz and kaolinite phases in the case of the sandstone. Simultaneously, peaks typical of quartz together with different silicate and aluminosilicate phases were observed in the old brick diffraction pattern.

No reflections which might be assigned to the silica crystalline phases in the pure silica layers were observed. Only in the case of the Aerosil$^{TM}$-derived thin film deposited on the titanium alloy can the elevated background in the range of 15°–30° [2θ] be observed, which is typical for an amorphous phase presence (Figure 6). It is very probable that most of the deposited thin films were obtained in the amorphous form, although this amorphous "halo" is not observed in their diffraction patterns because of the strong signal coming from the substrate. The amorphous character of the layers confirms their homogeneity and points to the isotropic character of the properties of the deposited coatings.

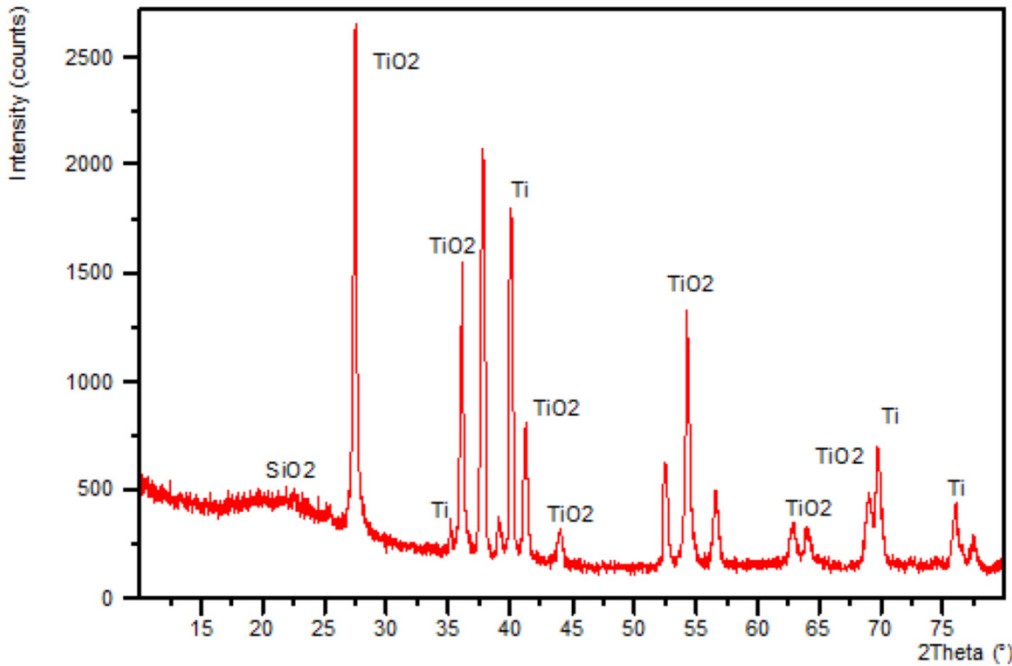

**Figure 6.** XRD diffraction pattern of the thin film deposited from Aerosil$^{TM}$ solution on the titanium alloy as the substrate.

### 3.3. SEM Studies of Samples

Coatings obtained on the porous materials (ceramics and natural stones) caused changes in the microstructure of the surface by creating thick, almost continuous layers, and in some cases also cube-shaped crystallites, rich in silicon and oxygen (according to the EDS spectra) crystallites (Figures 7 and 8).

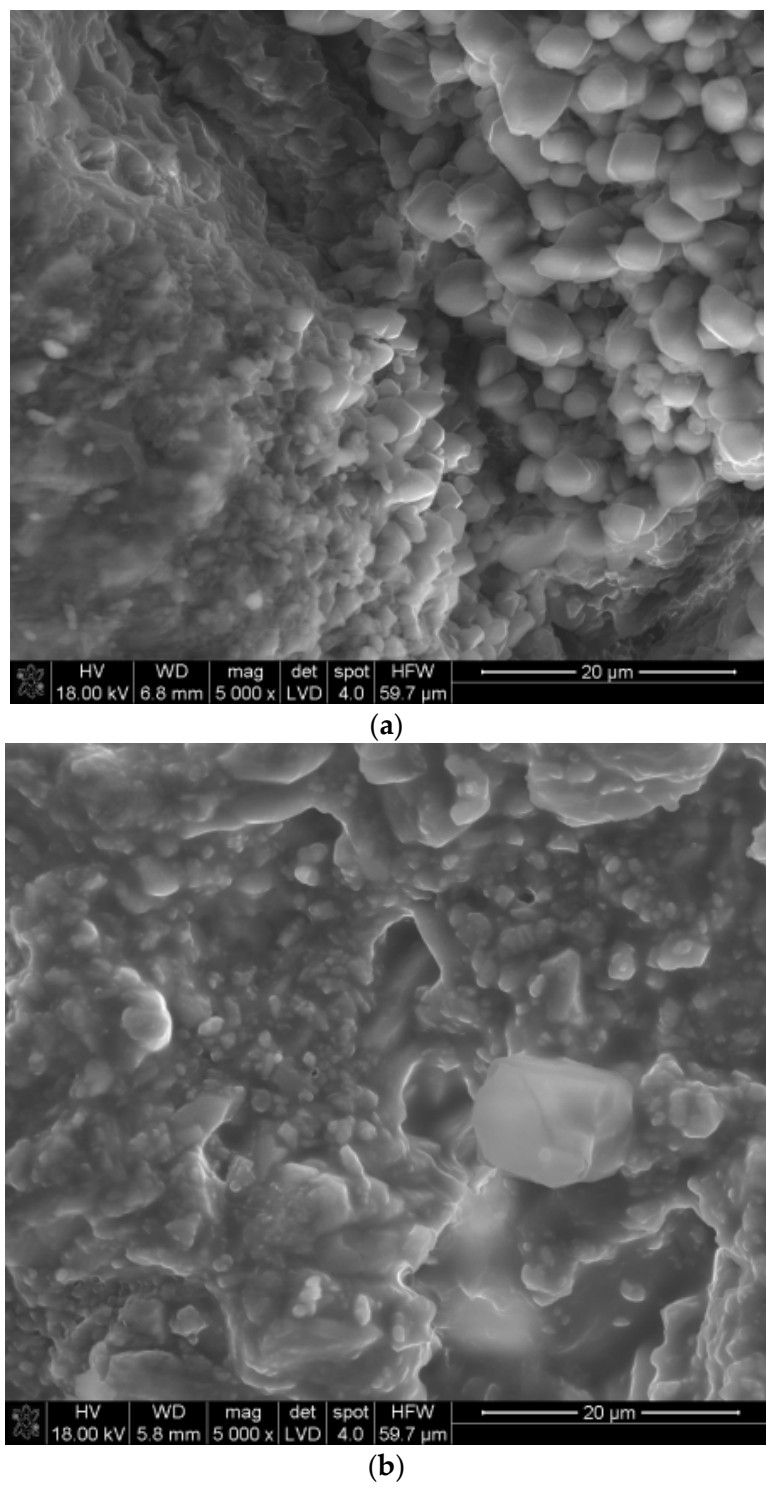

**Figure 7.** *Cont.*

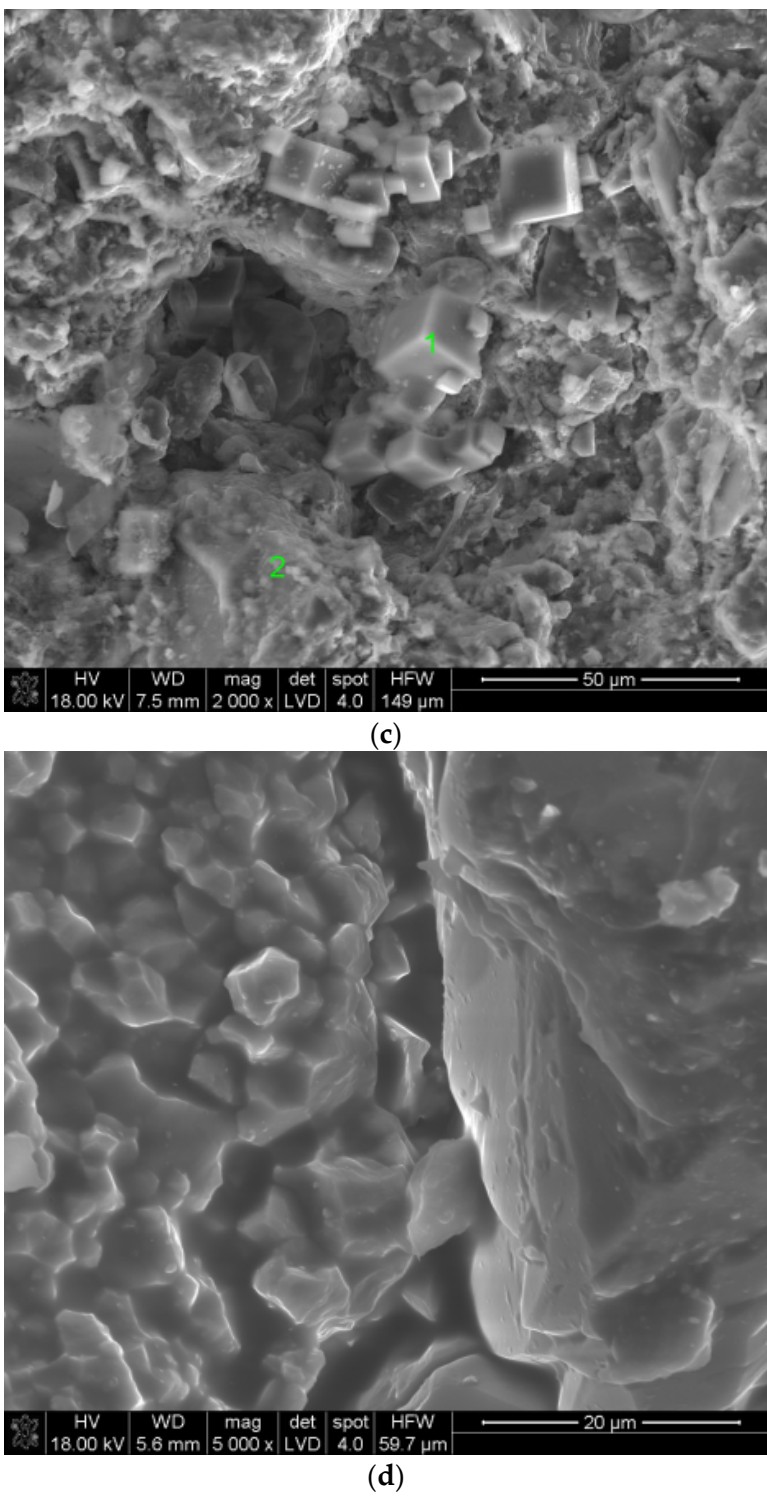

**Figure 7.** SEM images of: (**a**) limestone infiltrated with Ormosil IV; (**b**) limestone infiltrated with Sarsil H1 4/2; (**c**) old brick infiltrated with Sarsil H 14/2; (**d**) sandstone infiltrated with Ormocer V.

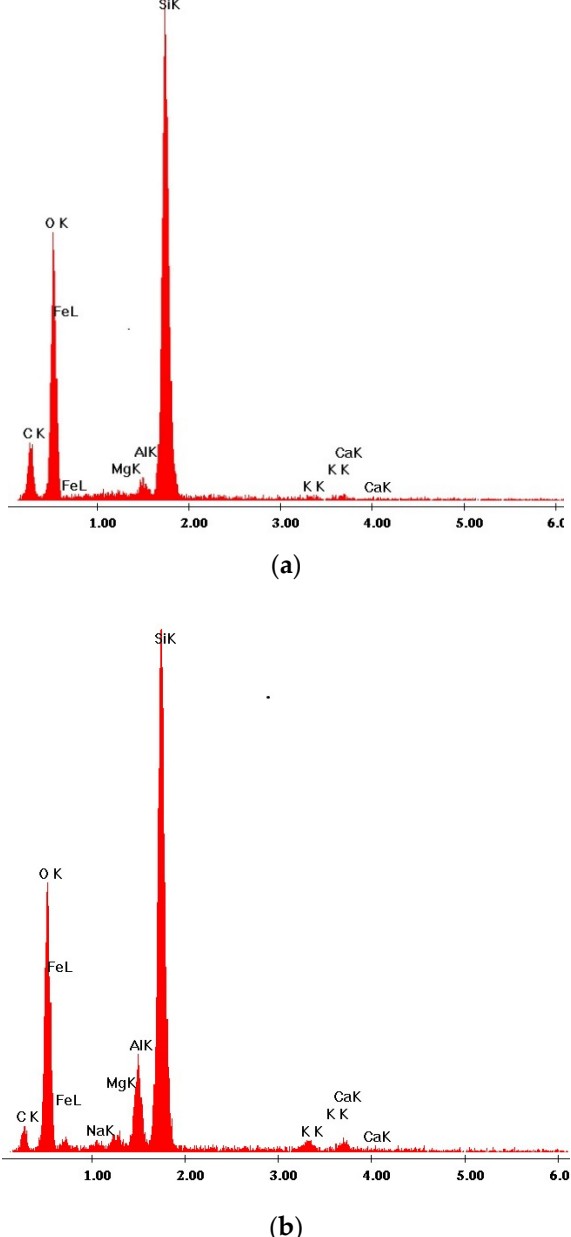

**Figure 8.** EDX microanalysis at points 1 (**a**) and 2 (**b**) on the surface of the old brick shown in Figure 7c.

Those rich in silica cubes were first observed on the surface of the old brick and sandstone in the cases of both the Sarsil H 14/2 preparation and the Ormocer-type sol applied as the infiltration solutions.

The surface morphology of thin films deposited on metallic substrates is determined to a greater extent by the annealing process than by the chemical composition of the sols used for the deposition. All images show the surfaces of coatings covered with a network of cracks, independent of the type of substrate and the sol applied for the deposition. Based on SEM images (Figures 9 and 10), one can draw the conclusion that all coatings present good adhesion and good tightness, probably due to deposition of multiple layers one by one on each sample surface.

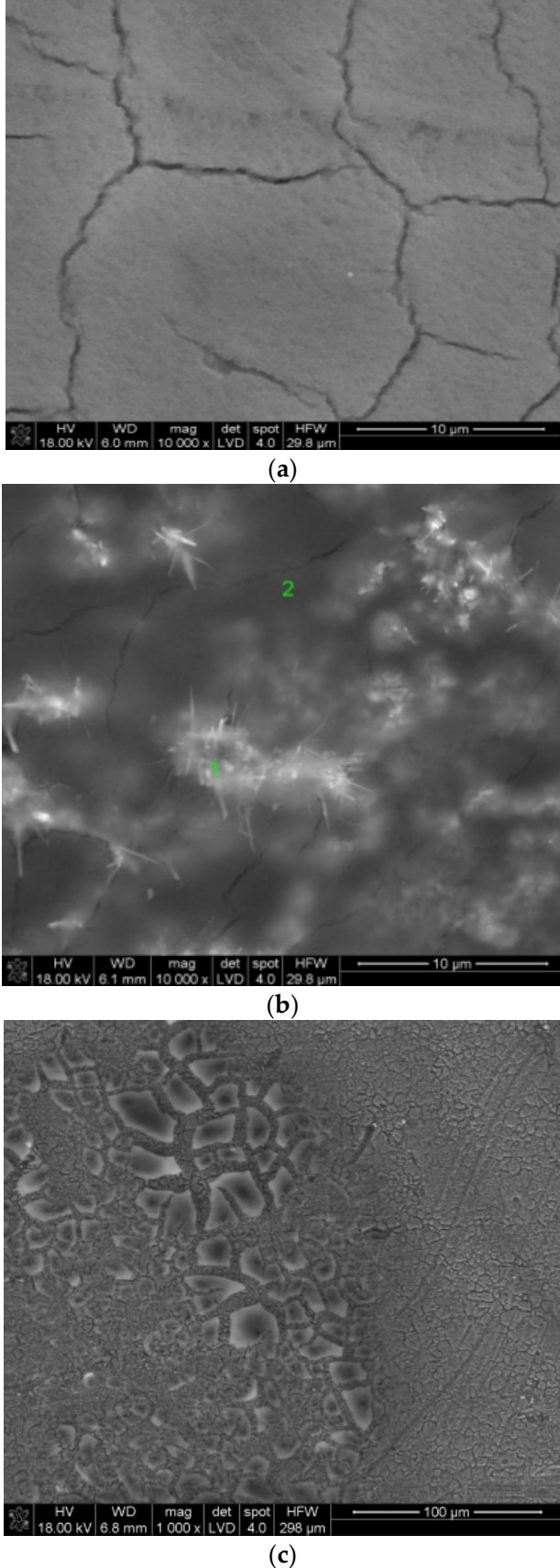

**Figure 9.** SEM images of: (**a**) the thin film deposited from Aerosil$^{TM}$ solution on steel; (**b**) the thin film deposited from Aerosil$^{TM}$ solution on titanium; (**c**) the thin film deposited from TEOS/DDS containing sol on steel substrate.

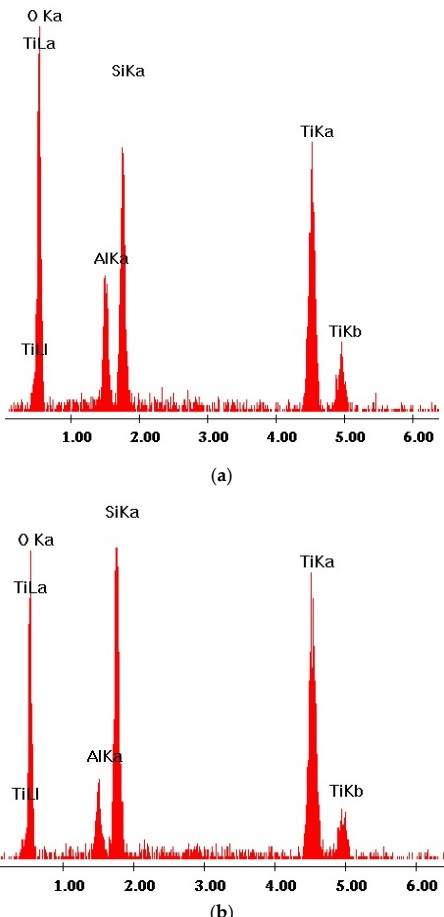

**Figure 10.** EDX microanalysis at points 1 (**a**) and 2 (**b**) on the surface of the thin film deposited from Aerosil$^{TM}$ solution on titanium and shown in Figure 9b.

### 3.4. AFM Studies of Samples

Images of the surfaces of thin films deposited on metallic substrates, obtained in the AFM microscope, allow observation of the topography of the samples in nanoscale in all three directions X, Y and Z, and estimation of the roughness of their surface. The roughness itself may have an influence on the different properties of the samples, as well as on their hydrophilic/hydrophobic properties. The AFM images were obtained only for thin films deposited on metallic substrates because of the very high surface roughness of ceramics or stones as substrates.

Roughness of the sample surface can be specified by the $R_a$ and $R_{max}$ indexes. $R_a$ (the arithmetic average roughness) is the most widely used roughness parameter, while $R_{max}$ describes the maximum distance between the highest and the lowest points on the studied area. These values ($R_a$ and $R_{max}$) were collected for the thin films deposited on metallic substrates and for non-coated clear substrates. First, it is easy to observe that $R_a$ and $R_{max}$ differed distinctly between the thin films and the uncoated samples (Table 1). In the case of thin films, the values collected increased simultaneously with the increasing concentration of silica in the sols applied for the coatings' deposition.

Based on the AFM images, the particle size of the selected thin films was estimated. From these images one can conclude that particle size seems to change with the increasing concentration of silica (from 5 to 14 wt.%) and varies between 20 and 200 nm. The strong influence of the type of $SiO_2$ precursor on particle size can also be observed. The biggest multiparticle aggregates are visible on the surface of the thin films deposited from Aerosil$^{TM}$ solution, while the smallest particles can be found in coatings deposited from the sol synthesized only with TEOS as the main component (Figure 11).

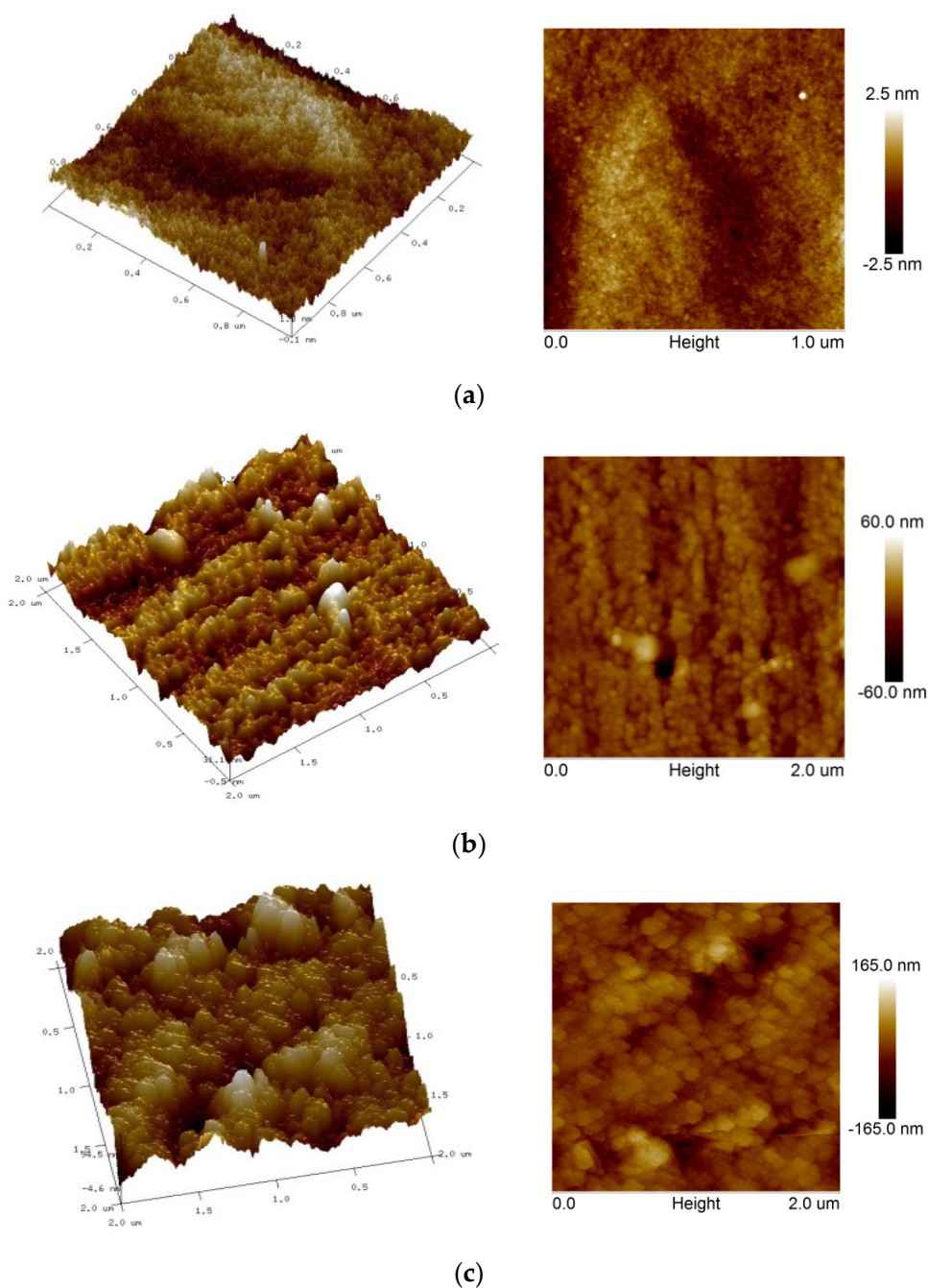

**Figure 11.** AFM 2D and 3D images of: (**a**) the thin film deposited from 5% SiO$_2$ sol (synthesized with TEOS) on steel; (**b**) the thin film deposited from TEOS/DDS containing sol on steel; (**c**) the thin film deposited from Aerosil$^{TM}$ solution on steel.

**Table 1.** The roughness $R_a$ and $R_{max}$ indexes of thin films deposited on steel and titanium substrates.

| No. | Sample/SiO$_2$ Precursor | Substrate | $R_{max}$ (nm) | $R_a$ (nm) |
|---|---|---|---|---|
| 1 | TEOS (14% SiO$_2$) | steel | 279 | 21.1 |
| 2 | TEOS (10% SiO$_2$) | steel | 152 | 20.3 |
| 3 | TEOS (5% SiO$_2$) | steel | 98 | 7.96 |
| 4 | TEOS/DDS | steel | 268 | 20.7 |
| 5 | Aerosil$^{TM}$ | steel | 226 | 19.1 |
| 6 | Aerosil$^{TM}$ | titanium | 161 | 17 |
| 7 | - | steel | 53.4 | 6.15 |
| 8 | - | steel | 70.9 | 10.2 |

*3.5. Wetting Angle Measurements*

The study of wetting angle measurements gives, in quite a simple way, important information on the hydrophilicity or hydrophobicity of the synthesized samples [27,28]. Wetting angle measurements were run for all synthesized materials and gave interesting results. Primarily, the attempts to obtain the wetting angle value of non-infiltrated porous mineral samples were undertaken and failed, except for the old brick, for which a set of measurements was collected which gave wetting angle values between 11.7° and 14.8°. Other materials behaved as highly hydrophilic sponges with very low values of wetting angle.

After the infiltration process, the measurement of wetting angles was run with success for all samples, which indicates the great influence on changing the properties of the samples' surfaces of all solutions applied for the deposition of layers (Tables 2–4). All obtained wetting angle values were between 107.4 and 144.6°. Because 90° is the boundary value of wetting angle, which allows for the division of materials into hydrophilic (below this value) or hydrophobic (above it), it can thus be concluded that all samples presented hydrophobic properties after the deposition of the coatings. These results also point to the great role played by the inclusion of methyl groups in the ability of the sols/solutions to improve or impart their hydrophobic properties to the different materials applied for the coating deposition.

**Table 2.** Wetting angle values of limestone.

| Sol/Preparation | Substrate | Average Value of Wetting Angle (°) |
|---|---|---|
| SILOXAN W 290 | limestone | 118.1 |
| SARSIL H/14/2 | limestone | 108.9 |
| ORMOCER V | limestone | 112.2 |
| ORMOCER IV | limestone | 115.8 |

**Table 3.** Wetting angle values of sandstone I.

| Sol/Preparation | Substrate | Average Value of Wetting Angle (°) |
|---|---|---|
| SILOXAN W 290 | sandstone I | 107.4 |
| SARSIL H/14/2 | sandstone I | 112.5 |
| ORMOCER V | sandstone I | 120.2 |
| ORMOCER IV | sandstone I | 118.0 |

**Table 4.** Wetting angle values of old brick.

| Sol/Preparation | Substrate | Average Value of Wetting Angle (°) |
|---|---|---|
| SILOXAN W 290 | old brick | 137.4 |
| SARSIL H/14/2 | old brick | 125.2 |
| ORMOCER V | old brick | 131.9 |
| ORMOCER IV | old brick | 144.6 |

The wetting angle measurements of the coatings deposited on metallic substrates gave quite different results. First of all, it was impossible to measure the wetting angle values for almost all layers despite the fact that some of them contained methyl groups. These samples behaved in the same way as porous samples before the infiltration and acted as sponges. The only exception was the thin film deposited on steel using sol containing 5% $SiO_2$ and prepared from TEOS as $SiO_2$ precursor. The wetting angle value of this sample reached 53.53°. This value is much smaller than 89.94° obtained for the pure, uncovered steel and also than the 65.74° value measured for the pure titanium substrate (Table 5).

**Table 5.** Wetting angle values of pure metallic substrates and selected thin films.

| Sol | Substrate | Average Value of Wetting Angle (°) |
|---|---|---|
| - | pure steel | 89.94 |
| - | pure titanium | 65.74 |
| TEOS (5% $SiO_2$) | steel | 53.53 |

The attempts to estimate the wetting angle values of the coatings on metallic substrates show that the influence of the surface topography after the annealing process on the hydrophilic/hydrophobic properties predominates over the influence of $CH_3$ groups incorporated into the structure of the selected thin films. The roughness of the samples with coatings is also much higher than that of non-covered samples (Table 1), which is probably caused by the annealing of samples at 500 °C. Such a process could cause the creation of a dense network of microcracks and thus make the samples hydrophilic due to the capillary lift of liquids through this network of microcracks.

## 4. Conclusions

In this work, two groups of coatings were obtained, which were deposited on metallic or porous substrates by the dip coating and the infiltration techniques, respectively. These coatings contained different $SiO_2$ precursors, which can be divided into two main sets. Precursors of the first set did not contain methyl groups as TEOS and Aerosil$^{TM}$. The second set consists of precursors containing $CH_3$ groups in their structure as diethoxy-dimethylsilane (DDS), applied in the research described. Thin films on metallic substrates were obtained from sols containing both types of $SiO_2$ precursors, while those deposited on porous mineral or ceramic substrates were obtained with sols or commercially available preparations containing precursors of the second set. Additionally, coatings deposited on metallic backgrounds were annealed at 500 °C in air or in argon (in the case of steel and titanium substrate, respectively).

FTIR spectra of all coatings containing methyl groups allowed to recognize the main band typical of $CH_3$ group vibrations at about 1289 $cm^{-1}$. As mentioned earlier in the text, those groups might be responsible for the hydrophobic properties of samples.

The very large differences between the wetting angle values of non-infiltrated and infiltrated porous samples point to the strong influence of the methyl groups present in sols and preparations applied for the infiltration on the hydrophobicity of samples. After the deposition of layers, samples became highly hydrophobic regardless of the evidently high roughness of their surface, visible in SEM pictures.

However, despite the presence of methyl groups in the structure of the selected and annealed thin films deposited on metallic substrates, the measurement of wetting angle values points to the much larger influence of the topography of the samples' surfaces on their high hydrophilic properties, as compared the presence of methyl groups incorporated into the selected thin films' structure and observed even after the annealing of the samples. This high hydrophilicity of the coatings on metallic substrates can probably be connected to the presence of a dense net of cracks that appeared after the annealing of the samples.

The main target of this work was to study the structure of the deposited coatings and the relation between the structure and the hydrophobic/hydrophilic properties of synthesized samples. The next step will be to carry out corrosion tests for uncoated and

coated samples. Combined with the previous results, this will allow for selection of the optimum thin film synthesis parameters (the manner of layer deposition and type of applied solution) in order to obtain the best possible corrosion protection for different types of materials such as ceramics, steel construction and natural stones and rocks.

**Funding:** This work was supported from the subsidy of the Ministry of Education and Science for the AGH University of Science and Technology in Kraków (Project No. 16.16.160.557).

**Institutional Review Board Statement:** Not applicable.

**Informed Consent Statement:** Not applicable.

**Data Availability Statement:** Data is contained within the article.

**Conflicts of Interest:** The authors declare no conflict of interest.

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
