# Peer review of "The Influence of Organically Modified Derivatives of Silica on the Structure and the Wetting Angle Values of Silica Coatings"

_coatings, doi:10.3390/coatings11091058_

Round 1

Reviewer 1 Report

Reviewers' comments:

Manuscript number: coatings-1345641

Title: “The influence of organically modified derivatives of silica on the structure and the wetting angle values of anticorrosive silica coatings”.

Comments: 

The manuscript reported “The influence of organically modified derivatives of silica on the structure and the wetting angle values of anticorrosive silica coatings”. The manuscript needs a detailed editing. It cannot be recommended for publication in the present form. I hope the following points would be helpful for the authors.

- The introduction is very poor and less informative. Authors should elaborate their introduction section by citing few more relevant references. The novelty of the work should also be highlighted.

- The experimental part must be detailed.

- Figure 3. FTIR spectra of the limestone – make clear.

- In part SEM: how the energy of the accelerator beam used?

- Figure 9, not clear make clear.

- The quality of EDX figures is too low.

- The authors are obliged to repeat the discussion part of the AFM studies of samples.

- The conclusion part should rebuild to let it fluent.

- Several faults: are added or missing spaces between words: see PDF file (for example – line number 44, 64, 66, 77, 118, 194, and etc..).

- References: And make all references in same format for volume number, page number and journal name.

So that I recommended this manuscript to major revision and for future process.  

Author Response

Response to the reviewer`s comments (reviewer 1):

  1. The introduction was rebuilt and some additional references were added to this section (lines 33-58 in the revised article).
  2. The experimential part was prepared in more detailed way – the part Result and discussion” (lines 60-119).
  3. Figure 3 together with Figure 2 ( the same type of picture) were prepared once more in much better resolution – they should be more clear now.
  4. The energy of accelerator beam varies between 10-18 kV during measurements (it was added together with the name of microscope)and is given at the bottom of every image, with HV abbreviation.
  5. All SEM images in the article were obtained with the highest possible resolution for this type of samples.
  6. The format of EDS pictures was changed from .tif to .jpg. It should make them being clearer.
  7. The discussion of results in AFM chapter was reapeated and changed (lines 222-242).
  8. The conclusions part was rewritten and should be more understable and fluent know
  9. The entire text was carefully checked to remove excess spaces or add missing ones.
  10. All references were checked and prepared according to the MPDI template.

Reviewer 2 Report

In this paper, the author prepared different sols by sol-gel method in order to obtain thin films by deposition onto various substrates.

For this paper, the answer is “Major Revision”. Please correct:

  • At Abstract, page 1, line 15, modify “thetraethylorthosilane” with “tetraethylorthosilane”
  • At Introduction, line 39, specify the abbreviations for CVD, PVD
  • At Section “2. Materials and methods”, line 54, the expression is wrong “All thin films were synthesized by the sol-gel method”. The thin films cannot be synthesized by sol-gel method. I suggest “All thin films were realized by deposition of sols synthesized by the sol-gel method and applying two techniques…”
  • Line 58, specify how the samples were cleaned.
  • At Section 2, for all instruments indicate the manufacture, city, country.
  • Mention how the samples were synthesized by sol-gel method.
  • At Section 3.1, write a conclusion of FTIR results.
  • Mention the corrosion tests.
  • In manuscript, don't write with bold the name of Figures (e.g., lines 125, 126, 129, 150).
  • Please, specify the possible application.
  • Correct the References using the Guide of the Journal.
  • Relevant reference can be included:

„Synthesis of Zinc Oxide Nanomaterials via Sol-Gel Process with Anti-Corrosive Effect for Cu, Al and Zn Metallic Substrates”, Coatings, 11, 2021, p.444

"Influence of Organically-Modified Montmorillonite and Synthesized Layered Silica Nanoparticles on the Properties of Polypropylene and Polyamide-6 Nanocomposites", Polymers 8(11), 2016, 386 

Author Response

Response to the reviewer`s comments (reviewer 2):

  1. At Abstract, I modyfied the word „thetraethylorthosilane” to „tetraethylorthosilane” (line 15).
  2. I specified the abbreviations for CVD and PVD (lines 45-46).
  3. I have changed the given phrase as suggested:”… All thin films were realized by the deposition of sols synthesized by the sol-gel method and applying two techniques:…” (lines 60-61)
  4. I specified the procedures of samples cleaning (lines 64-70)
  5. At section 2, I specyfied each instrument with manufacture, city and country (lines 106-119)
  6. The description of samples synthesis by sol-gel method was given in more detailed way in the section „Materials and methods” (lines 81-90).
  7. At section 3.1. I changed the whole part concerning FTIR studies and added some conclusions (lines 122-174)
  8. The corrosion tests are predicted in the future (line 318). The first step of my research was to investigate the structure of synthesized coatings and find the relation between the structure and the hydrophilic/hydrophobic properties of samples.
  9. All bold letters were removed from Figures captions.
  10. The possibile applications are mentioned in the introduction and the conclusions parts (lines 36-38, 319-322)
  11. The references were corrected according to the Guide of MPDI Journals.
  12. The suggested references werre addedto the refences list (positions no 10 and 11)

Round 2

Reviewer 1 Report

Reviewers' comments:

The authors revised the manuscript according to the reviewers' comments.

Author Response

Thank you very much for your valuable comments which helped to improve this article.

Reviewer 2 Report

Dear Sirs,

The authors made the recommended changes. In order to be publish, please make the changes:

  • Because the authors don’t made the corrosion tests, I suggest to change the title of paper and remove the “anticorrosive” name
  • At Abstract, modify “synthesized coatings” with “prepared coatings”
  • At Section 2, modify “dimethyldiethxyosilane” with “dimethyldiethoxysilane”

Author Response

Answers on reviewer`s comments:

  1. The title was changed according to the reviewer`s suggestion.
  2. The phrase: "synthesized coatings" was substituted by "prepared coatings" in section "Abstract".
  3. The letter "o" was put in a proper position in the word "dimethyldiethoxysilane" in Section 2.